# DIFFUSION-SDPO: SAFEGUARDED DIRECT PREFERENCE OPTIMIZATION FOR DIFFUSION MODELS

## ABSTRACT

Text-to-image diffusion models deliver high-quality images, yet aligning them with human preferences remains challenging. We revisit diffusion-based Direct Preference Optimization (DPO) for these models and identify a critical pathology: enlarging the preference margin does not necessarily improve generation quality. In particular, the standard Diffusion-DPO objective can increase the reconstruction error of both winner and loser branches. Consequently, degradation of the less-preferred outputs can become sufficiently severe that the preferred branch is also adversely affected even as the margin grows. To address this, we introduce Diffusion-SDPO, a safeguarded update rule that preserves the winner by adaptively scaling the loser gradient according to its alignment with the winner gradient. A first-order analysis yields a closed-form scaling coefficient that guarantees the error of the preferred output is non-increasing at each optimization step. Our method is simple, model-agnostic, broadly compatible with existing DPO-style alignment frameworks and adds only marginal computational overhead. Across standard text-to-image benchmarks, Diffusion-SDPO delivers consistent gains over preference-learning baselines on automated preference, aesthetic, and prompt alignment metrics.

## 1 INTRODUCTION

Text-to-image diffusion models Croitoru et al. (2023) have achieved remarkable success in generating diverse and high-quality images Labs (2024); Google (2025). However, aligning these powerful generative models with nuanced human preferences remains a critical challenge. Recent approaches have begun to incorporate human feedback Christiano et al. (2017) into diffusion model training, drawing inspiration from alignment techniques used in large language models. In particular, Direct Preference Optimization (DPO) Rafailov et al. (2023) has emerged as a promising alternative to reinforcement learning for finetuning on human preferences. DPO directly optimizes the model on pairwise human comparisons (winner vs. loser outputs), and has been successfully adapted to text-to-image diffusion models in methods Wallace et al. (2024); Hong et al. (2025); Zhu et al. (2025); Li et al. (2025a;b) to improve visual appeal and prompt alignment. Despite these advances, we find that existing DPO-based alignment of diffusion models still faces a fundamental limitation: simply maximizing the preference margin between "winner" and "loser" outputs does *not* necessarily translate to better absolute generation quality of the finetuned model.

In our empirical analysis, we find that standard Diffusion-DPO Wallace et al. (2024) exhibits unstable training dynamics, and the model's generative quality can deteriorate as training proceeds. As illustrated in the left part of Fig. 1, we find that both the winner's and loser's denoising losses tend to increase over time, even though the preference margin ($\mathcal{L}^w - \mathcal{L}^l$) becomes more negative in the intended direction. This indicates that the model is widening the relative preference gap by making the less-preferred outputs worse, rather than truly improving the preferred outputs. In other words, relative alignment comes at the expense of absolute quality. The lack of a safeguard on the winner's loss in existing DPO objectives leads to unstable training and potential collapse, corroborating observations in prior work Pal et al. (2024); Xiao et al. (2024); Shekhar et al. (2025) that overly aggressive preference optimization can harm generative performance. These findings motivate the need for a new approach to preference-based diffusion finetuning that can increase preference alignment while preserving or improving the quality of the preferred outputs.

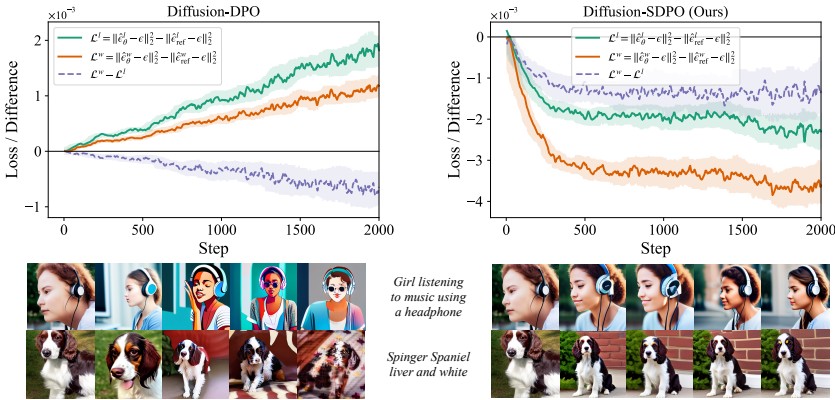

Figure 1: Training dynamics of preference losses during DPO finetuning *without (left)* and *with (right)* our safe-$\lambda$ mechanism on SD 1.5 Rombach et al. (2022). Images beneath the plots illustrate samples generated at training steps $\{0, 500, 1000, 1500, 2000\}$.

To address this challenge, we propose Diffusion-SDPO[1] – a Safeguarded Direct Preference Optimization method for diffusion models. The key idea in Diffusion-SDPO is to introduce a simple yet effective winner-preserving update rule that controls the influence of the loser sample's gradient at each training step. In contrast to standard DPO Rafailov et al. (2023); Wallace et al. (2024) which updates the model by contrasting winner and loser equally, we derive an adaptive scaling factor for the loser's gradient based on the geometry of the winner and loser gradients. Intuitively, our method downweights the loser branch's contribution whenever its gradient is misaligned with the winner's gradient. Grounded in a first-order analysis, the safeguard computes a closed-form $\lambda_{\text{safe}}$ from the inner product of the winner and loser gradients, guaranteeing that each step does not worsen the preferred output's reconstruction loss. In practice, Diffusion-SDPO seamlessly modifies the DPO objective with this adaptive loser scaling (see Fig. 1, right), which expands the preference margin while strictly controlling the absolute error of preferred outputs. Notably, our approach is model-agnostic and can be applied on top of various diffusion alignment frameworks Wallace et al. (2024); Hong et al. (2025); Li et al. (2025a); Zhu et al. (2025), acting as a plug-in optimizer that stabilizes training. Our contributions can be summarized as follows:

- We show that enlarging the winner–loser margin in diffusion preference optimization does not guarantee higher quality and can degrade preferred outputs, revealing a gap between relative alignment and absolute error control.

- Based on these analysis, we propose *Diffusion-SDPO*, a winner-preserving training scheme that adaptively scales the loser gradient by its geometric alignment with the winner gradient to first order. Our method is simple to implement and adds negligible overhead.

- Extensive experiments on SD 1.5 Rombach et al. (2022), SDXL Podell et al. (2024), and the industrial-scale Ovis-U1 Wang et al. (2025a) show that our method consistently improves preference metrics while preserving or enhancing aesthetic quality, stabilizing training, and avoiding collapse. Moreover, the gains hold across text-to-image models, unified generators, and image-editing setups.

## 2 RELATED WORK.

**Diffusion Models for Text-to-Image and Unified Generation.** Diffusion models have become a leading paradigm for image synthesis, offering strong quality and diversity Croitoru et al. (2023). Denoising diffusion with a variational objective Ho et al. (2020) and continuous-time score-based

---

[1]Throughout the text, "Diffusion-SDPO" is used as a conceptual umbrella for our method and its guiding principles. When referring to concrete instantiations, we write "X+SDPO" to denote the integration of SDPO with a specific base DPO variant $X$ (e.g., Diffusion-DPO, DSPO, DMPO), which clarifies the application setting and configuration.

formulations with SDEs Jo et al. (2022); Song et al. (2022) underpin modern systems. Refinements such as EDM Karras et al. (2022) and rectified flow or flow matching Liu et al. (2023); Lipman et al. (2023) clarify objectives and improve robustness. Guidance-based conditioning Dhariwal & Nichol (2021); Ho & Salimans (2022) enhances controllability. For text-to-image generation, latent diffusion Rombach et al. (2022) enables efficient high-resolution synthesis and supports large systems like SD3 Esser et al. (2024) and FLUX Labs (2024). In parallel, unified generators handle text-to-image and image editing within a single model Wang et al. (2025a). Our method applies to both families and is architecture-agnostic, working with UNet Ronneberger et al. (2015)-style and DiT Peebles & Xie (2023)-style backbones.

**Preference Optimization for Diffusion Models.** Direct Preference Optimization Rafailov et al. (2023); Wallace et al. (2024); Gambashidze et al. (2024) has been adapted to diffusion models to align generation with human comparisons while avoiding full reinforcement learning. A broad class of variants Wang et al. (2025b); Hong et al. (2025) calibrates the preference margin or the relative branch influence to improve stability and protect the generation. Other approaches seek to guide the update directions and step magnitudes in LLMs Zhao et al. (2023) by employing subspace projections and modest objective clipping Yang et al. (2024); Cho et al. (2025); Xiao et al. (2024); Chowdhury et al. (2024); Huang et al. (2025). Related work such as DPOP Pal et al. (2024) promotes positivity constraints to mitigate failure modes in preference optimization, and MaPPO Lan et al. (2025) incorporates prior knowledge via a maximum-a-posteriori objective. Diffusion-specific methods further account for the multi-step nature of denoising by reweighting across timesteps or by adding entropy regularization, exemplified by Balanced-DPO Tamboli et al. (2025), DSPO Zhu et al. (2025), and SEE-DPO Shekhar et al. (2025). In contrast, our Diffusion-SDPO introduces a per-step, geometry-aware safe scaling factor based on the inner product between winner and loser output-space gradients, which provides direct control over the winner loss at each step while continuing to expand the preference margin.

# 3 PRELIMINARIES

**Diffusion Models.** Diffusion models Sohl-Dickstein et al. (2015); Ho et al. (2020) construct a Markov chain that gradually corrupts clean data with additive noise and then learn a parametric denoiser to invert this corruption. Let a variance schedule $\{\beta_t\}_{t=1}^{T}$ be given and define $\alpha_t = 1 - \beta_t$ and $\bar{\alpha}_t = \prod_{s=1}^{t} \alpha_s$. The forward process can be defined as:

$$q(x_t \mid x_{t-1}) = \mathcal{N}\big(x_t; \sqrt{\alpha_t}\, x_{t-1}, (1 - \alpha_t)\,\mathbf{I}\big), \tag{1}$$

which implies the following closed-form perturbation of a clean sample $x_0$:

$$x_t = \sqrt{\bar{\alpha}_t}\, x_0 + \sqrt{1 - \bar{\alpha}_t}\, \epsilon, \qquad \epsilon \sim \mathcal{N}(\mathbf{0}, \mathbf{I}). \tag{2}$$

Equivalently, the marginal distribution conditioned on $x_0$ is

$$q(x_t \mid x_0) = \mathcal{N}\big(x_t; \sqrt{\bar{\alpha}_t}\, x_0, (1 - \bar{\alpha}_t)\,\mathbf{I}\big). \tag{3}$$

Learning proceeds by training a network $\epsilon_\theta$ that receives the noised input $x_t$ and the time index $t$ to predict the injected noise. Using the reparameterization in Eq. 2, the standard objective minimizes mean squared error between the true noise and the prediction:

$$\mathcal{L}_{\text{diffusion}} = \mathbb{E}_{x_0 \sim p_{\text{data}},\ t \sim \text{Uniform}\{1,\ldots,T\},\ \epsilon \sim \mathcal{N}(\mathbf{0},\mathbf{I})} \big\| \epsilon_\theta\big(x_t, t\big) - \epsilon \big\|_2^2. \tag{4}$$

Minimizing $\mathcal{L}_{\text{diffusion}}$ yields a time-aware denoiser that can be applied in reverse order to iteratively remove noise and synthesize new samples from an initial Gaussian latent.

**Diffusion Model Alignment via Preference.** Given a prompt $c$ and two images $x_0^w$ (preferred, "winner") and $x_0^l$ (less preferred, "loser"), preference alignment for diffusion models seeks parameters $\theta$ such that the model assigns higher likelihood to $x_0^w$ than to $x_0^l$ Gambashidze et al. (2024); Wallace et al. (2024). A diffusion sampler produces a trajectory $(x_T, \ldots, x_0)$ and, at each time $t$, a reverse conditional $p_\theta(x_t \mid x_{t+1}, c)$ Ho et al. (2020); Sohl-Dickstein et al. (2015); Liu et al. (2023). To instantiate DPO in this setting, we adopt the standard formulation wherein the stepwise preference score is the log-likelihood ratio with respect to a frozen reference model Wallace et al. (2024); Zhu et al. (2025):

$$r_t(x_t, c) = \beta \log \frac{p_\theta(x_t \mid x_{t+1}, c)}{p_{\text{ref}}(x_t \mid x_{t+1}, c)}. \tag{5}$$

The Diffusion–DPO Wallace et al. (2024) loss applies Bradley-Terry-style Bradley & Terry (1952) logistic regression to the winner-loser pair at the same $t$:

$$\mathcal{L}_{\text{Diffusion-DPO}} \; = \; - \, \mathbb{E}\Big[ \log \sigma\Big( r_t(x_t^w, c) \; - \; r_t(x_t^l, c) \Big) \Big], \tag{6}$$

and averages Eq. 6 over $t \in \{0, \ldots, T-1\}$ (or samples a single $t$ per pair for an unbiased stochastic estimator). Equivalently, substituting Eq. 5 into Eq. 6 gives the explicit form

$$\mathcal{L}_{\text{Diffusion-DPO}} = - \, \mathbb{E}\left[ \log \sigma\left( \beta \log \frac{p_\theta(x_t^w \mid x_{t+1}^w, c)}{p_{\text{ref}}(x_t^w \mid x_{t+1}^w, c)} - \beta \log \frac{p_\theta(x_t^l \mid x_{t+1}^l, c)}{p_{\text{ref}}(x_t^l \mid x_{t+1}^l, c)} \right) \right]. \tag{7}$$

Under common parameterizations, Eq. 5 reduces to simple residual comparisons. For DDPM-style Gaussians Ho et al. (2020); Sohl-Dickstein et al. (2015), writing $\hat{\epsilon}_\theta = \epsilon_\theta(x_{t+1}, c, t)$ for the predicted noise, $\hat{\epsilon}_{\text{ref}} = \epsilon_{\text{ref}}(x_{t+1}, c, t)$ for the reference noise and $\epsilon$ for the ground-truth noise that forms $x_t$, the log-ratio can be expressed as:

$$\log \frac{p_\theta(x_t \mid x_{t+1}, c)}{p_{\text{ref}}(x_t \mid x_{t+1}, c)} \propto -\tfrac{1}{2}\big\|\hat{\epsilon}_\theta - \epsilon\big\|_2^2 + \tfrac{1}{2}\big\|\hat{\epsilon}_{\text{ref}} - \epsilon\big\|_2^2 + \text{const}, \tag{8}$$

and an analogous expression holds for velocity or flow-matching parameterizations by replacing the noise residual with the corresponding target Liu et al. (2023). For notational brevity, we write the stepwise contrastive objective as $\mathcal{L}(x_{t+1}, c, t) = \tfrac{1}{2}\|\epsilon_\theta(x_{t+1}, c, t) - \epsilon\|_2^2 - \tfrac{1}{2}\|\epsilon_{\text{ref}}(x_{t+1}, c, t) - \epsilon\|_2^2$. Hence, the winner and loser margin loss are defined as $\mathcal{L}^w = \mathcal{L}(x_{t+1}^w, c, t)$ and $\mathcal{L}^l = \mathcal{L}(x_{t+1}^l, c, t)$, respectively. Substituting Eq. 8 into Eq. 7 gives the training loss

$$\hat{\mathcal{L}}_{\text{Diffusion-DPO}} = - \, \mathbb{E}_{t \sim \text{Uniform}\{0,\ldots,T-1\}, \epsilon \sim \mathcal{N}(\mathbf{0}, \mathbf{I}), (c, x_0^w, x_0^l) \sim \mathcal{D}}\Big[ \log \sigma\Big( - \beta \left( \mathcal{L}^w - \mathcal{L}^l \right) \Big) \Big], \tag{9}$$

where $\mathcal{D} = \{(c, x_0^w, x_0^l)\}$ denotes the DPO training dataset.

**Limitations of Standard DPO.** Substituting Eq. 8 into Eq. 7 yields an implementable objective whose inner term is the per-step error difference between winner and loser branches. Diffusion–DPO Wallace et al. (2024) thus encourages decreasing the winner's prediction error while increasing the loser's at the same timestep. However, this objective does not guarantee a monotonic decrease of the winner loss. Empirically, over-penalizing the loser can also worsen the preferred sample. In the left part of Fig. 1, the margin $\mathcal{L}^w - \mathcal{L}^l$ becomes increasingly negative, yet both $\mathcal{L}^w$ and $\mathcal{L}^l$ increase, indicating degradation of absolute performance and potential instability or collapse. This exposes a gap between *relative* alignment (widening the margin) and *absolute* error control (preserving the preferred sample). The difficulty is that the winner and loser gradients are misaligned and vary across timesteps. We therefore introduce a simple stepwise update that, to first order, guarantees the preferred loss does not increase at each step while still promoting margin expansion.

## 4 METHOD: DIFFUSION-SDPO (SAFE DPO)

We propose **Diffusion-SDPO**, a novel preference optimization scheme that adds a safety guard to the DPO update. The method adaptively scales the influence of the loser branch by a time-dependent factor $\lambda_t$ so that the preferred sample's loss $\mathcal{L}^w$ does not increase after each parameter update. In practice, we follow the standard Diffusion–DPO pipeline: given a prompt $c$ and a pair $(x_0^w, x_0^l)$, we compute the per-sample losses $\mathcal{L}^w$ and $\mathcal{L}^l$ at the same diffusion time $t$, and then modify the backpropagated update by multiplying the loser-branch gradient by the safety factor to enforce a safe update condition. This directly addresses the limitation discussed above, because preventing any increase in the preferred loss ensures that preference-driven updates do not degrade the preferred output while still improving the preference margin.

### 4.1 SAFE UPDATE VIA FIRST-ORDER APPROXIMATION

Our objective is to ensure that a gradient update driven by the preference loss (cf. Eq. 7) does not increase the winner's loss. For clarity of exposition, consider a linearized preference objective

combining the two branches:[2]

$$\mathcal{L}^{\text{pref}}(\theta) = \mathcal{L}^w(\theta) - \lambda \cdot \mathcal{L}^l(\theta), \tag{10}$$

where $\lambda > 0$ is a scalar that adjusts the relative weight on the loser's loss. Setting $\lambda = 1$ recovers the intuitive gradient direction of standard DPO (decrease $\mathcal{L}^w$, increase $\mathcal{L}^l$), while $\lambda > 1$ would place even more emphasis on penalizing the loser. Our goal is to find an upper bound on $\lambda$ that guarantees $\mathcal{L}^w$ will not increase for an infinitesimal gradient step on $\mathcal{L}^{\text{pref}}$.

Let $\nabla_\theta \mathcal{L}^w$ and $\nabla_\theta \mathcal{L}^l$ denote the gradients of the winner and loser losses, respectively. A gradient descent step of size $\eta$ on Eq. 10 gives the parameter update:

$$\Delta\theta = -\eta \cdot \nabla_\theta \mathcal{L}^{\text{pref}} = -\eta\Big(\nabla_\theta \mathcal{L}^w - \lambda \nabla_\theta \mathcal{L}^l\Big). \tag{11}$$

The first-order change in the winner's loss can be approximated by a Taylor expansion:

$$\Delta\mathcal{L}^w \approx \nabla_\theta \mathcal{L}^{w\top} \Delta\theta = -\eta\Big(\|\nabla_\theta \mathcal{L}^w\|_2^2 - \lambda \nabla_\theta \mathcal{L}^{w\top} \nabla_\theta \mathcal{L}^l\Big). \tag{12}$$

To *prevent* increase in $\mathcal{L}^w$, we require $\Delta\mathcal{L}^w \leq 0$, i.e., $\nabla_\theta \mathcal{L}^{w\top} \Delta\theta \leq 0$. Ignoring the trivial positive factor $\eta$, the safety condition becomes:

$$\|\nabla_\theta \mathcal{L}^w\|_2^2 - \lambda \nabla_\theta \mathcal{L}^{w\top} \nabla_\theta \mathcal{L}^l \geq 0. \tag{13}$$

Solving for $\lambda$ yields a bound on the allowable loser weight:

$$\lambda \leq \frac{\|\nabla_\theta \mathcal{L}^w\|_2^2}{\nabla_\theta \mathcal{L}^{w\top} \nabla_\theta \mathcal{L}^l}. \tag{14}$$

Notably, if the dot product $\nabla_\theta \mathcal{L}^{w\top} \nabla_\theta \mathcal{L}^l$ is negative or zero, then Eq. 13 is automatically satisfied for any $\lambda \geq 0$. In those cases, the update is intrinsically safe: the loser branch either helps reduce $\mathcal{L}^w$ or affects orthogonal parameter directions. The problematic scenario is when $\nabla_\theta \mathcal{L}^{w\top} \nabla_\theta \mathcal{L}^l > 0$, i.e., the loser's gradient has a component that would raise the winner's loss. Eq. 14 then yields a finite positive $\lambda$ threshold. Any choice of $\lambda$ above this threshold would violate the safety inequality, leading to $\Delta\mathcal{L}^w > 0$ to first order. Conversely, choosing $\lambda$ at or below this threshold ensures $\Delta\mathcal{L}^w \approx 0$ or negative, guaranteeing that the winner's loss does not increase.

### 4.2 Closed-Form Safeguard in Output Space

Directly evaluating the parameter-space bound in Eq. 14 is infeasible for a high-dimensional model, since it would require computing and storing the full gradients $\nabla_\theta \mathcal{L}^w$ and $\nabla_\theta \mathcal{L}^l$ just to take their dot product. However, we can derive a convenient proxy by considering gradients in the model's *output space*. Modern diffusion models predict a noise or image tensor as output, and the training loss (e.g., a denoising score-matching loss Ho et al. (2020)) is defined on this output. Let $o^w$ and $o^l$ denote the model's output activations for the winner and loser branches respectively (for example, $o$ could be the predicted noise residual at a certain diffusion step). Using the chain rule, we have $\nabla_\theta \mathcal{L}^w = J^{w\top} \nabla_o \mathcal{L}^w$ and $\nabla_\theta \mathcal{L}^l = J^{l\top} \nabla_o \mathcal{L}^l$, where $J$ is the Jacobian $\partial o/\partial \theta$ and $\nabla_o \mathcal{L}$ is the gradient of the loss with respect to the model output. The term $\nabla_\theta \mathcal{L}^{w\top} \nabla_\theta \mathcal{L}^l$ can then be written as:

$$\nabla_\theta \mathcal{L}^{w\top} \nabla_\theta \mathcal{L}^l = (\nabla_o \mathcal{L}^w)^\top \big(J^w J^{l\top}\big)(\nabla_o \mathcal{L}^l). \tag{15}$$

If we assume a local near-isometry, namely $J^w J^{l\top} \approx I$ (see Sec. A for more details), then the parameter-space inner product is well approximated by the Euclidean dot product in output space, $(\nabla_o \mathcal{L}^w)^\top (\nabla_o \mathcal{L}^l)$. Let $g^w = \nabla_o \mathcal{L}^w$ and $g^l = \nabla_o \mathcal{L}^l$ denote the output-space gradients for the winner and the loser. We then define the safe step-size as:

$$\lambda_{\text{safe}} := \frac{\|g^w\|_2^2}{g^{w\top} g^l}, \tag{16}$$

---

[2] In practice, the actual Diffusion-DPO gradient (Eq. 6) includes a logistic scaling factor $\sigma(\cdot)$ that multiplies the winner and loser gradients equally, thus not altering the update direction. We therefore analyze the simpler weighted difference objective that captures the same first-order direction.

---

**Algorithm 1: Diffusion-SDPO**: Winner-preserving scaling for DPO-style diffusion training

---

**Input:** DPO dataset $\mathcal{D} = \{(c, x_0^w, x_0^l)\}$; model $\epsilon_\theta$; reference $\epsilon_{\text{ref}}$; safety slack $\mu \in [0, 1]$;
        schedule length (timesteps) $T$; learning rate $\eta$.

**while** *not converged* **do**

    1. Sample $t \sim \text{Uniform}\{0, \ldots, T-1\}, \epsilon \sim \mathcal{N}(\mathbf{0}, \mathbf{I}), (c, x_0^w, x_0^l) \sim \mathcal{D}$.

    2. Get $(x_{t+1}^w, x_{t+1}^l)$ from Eq. 2 and compute

    $\hat{\epsilon}_\theta^w = \epsilon_\theta(x_{t+1}^w, c, t), \hat{\epsilon}_\theta^l = \epsilon_\theta(x_{t+1}^l, c, t), \hat{\epsilon}_{\text{ref}}^w = \epsilon_{\text{ref}}(x_{t+1}^w, c, t), \hat{\epsilon}_{\text{ref}}^l = \epsilon_{\text{ref}}(x_{t+1}^l, c, t)$.

    3. Get per-branch residual objectives:

$$\mathcal{L}^w = \tfrac{1}{2}\|\hat{\epsilon}_\theta^w - \epsilon\|_2^2 - \tfrac{1}{2}\|\hat{\epsilon}_{\text{ref}}^w - \epsilon\|_2^2, \quad \mathcal{L}^l = \tfrac{1}{2}\|\hat{\epsilon}_\theta^l - \epsilon\|_2^2 - \tfrac{1}{2}\|\hat{\epsilon}_{\text{ref}}^l - \epsilon\|_2^2.$$

    4. Compute $\lambda_{\text{safe}}$ using Eq. 17: $\lambda_{\text{safe}} = (1 - \mu)\|g^w\|_2^2 / g^{w\top}g^l$.

    5. Scale only loser gradients: $\mathcal{L}_{\text{scaled}}^l = \mathcal{L}_{\text{detach}}^l + \lambda_{\text{safe}}(\mathcal{L}^l - \mathcal{L}_{\text{detach}}^l)$.

      Mark: $\mathcal{L}_{\text{detach}}^l$ is a copy of $\mathcal{L}^l$ with gradients detached (no gradient flow).

    6. Build loss using Eq. 9: $\mathcal{L}_{\text{DPO}} = -\log\sigma\left(-\beta(\mathcal{L}^w - \mathcal{L}_{\text{scaled}}^l)\right)$.

    7. Update $\theta \leftarrow \theta - \eta\,\nabla_\theta\,\mathcal{L}_{\text{DPO}}$.

**Output:** Finetuned model $\epsilon_\theta$.

---

if $g^{w\top}g^l > 0$, and we set $\lambda_{\text{safe}} = +\infty$ (i.e., impose no cap on $\lambda$) if $g^{w\top}g^l \leq 0$. In words, $\lambda_{\text{safe}}$ is a closed-form upper bound on the loser weight *based on output-space gradients*. It is computed cheaply per batch by taking the dot product of the two model-output error signals. Whenever the loser's error vector has a positive correlation with the winner's error vector ($g^{w\top}g^l > 0$), $\lambda_{\text{safe}}$ provides a finite limit to how strongly we can apply the loser's gradient without risking an increase in the winner's loss. On the other hand, if the errors are orthogonal or negatively correlated ($g^{w\top}g^l \leq 0$), the winner is not threatened by the loser's update.

### 4.3 Training with Diffusion-SDPO

At each iteration, we use preference pairs with a shared prompt $c$ and diffusion step $t$. Two forward passes yield the per-step losses $(\mathcal{L}^w, \mathcal{L}^l)$ and the corresponding output-space gradients $(g^w, g^l)$. We set the safeguard

$$\lambda_{\text{safe}} = \frac{(1 - \mu)\,\|g^w\|_2^2}{g^{w\top}g^l}, \tag{17}$$

and clip it to $[0, 1]$ for stability (if $g^{w\top}g^l \leq 0$, we set $\lambda_{\text{safe}} = 1$). Here $\mu \in [0, 1]$ is a safety slack to offset curvature and mini-batch noise in the first-order estimate (see Sec. B for more details). Any $\lambda < \lambda_{\text{safe}}$ ensures $\|\nabla_\theta\mathcal{L}^w\|_2^2 - \lambda\,\nabla_\theta\mathcal{L}^{w\top}\nabla_\theta\mathcal{L}^l > 0$, hence $\Delta\mathcal{L}^w < 0$ and the winner loss strictly decreases. For the logistic DPO objective, we implement this by scaling the backpropagated loser gradient with $\lambda_{\text{safe}}$. Algorithm 1 summarizes the procedure to integrate SDPO into Diffusion-DPO.

## 5 Experiments

### 5.1 Experimental Setting

**Datasets and Models.** We finetune Stable Diffusion 1.5 (SD1.5) and SDXL on preference pairs from Pick-a-Pic V2 (Pick V2) Kirstain et al. (2023) training set. For evaluation, we use the test prompts from Pick V2, HPS V2 Wu et al. (2023), and PartiPrompts Yu et al. (2022) following Zhu et al. (2025); Li et al. (2025a). Beyond SD1.5 and SDXL, we also conduct experiments on Ovis-U1 Wang et al. (2025a) (3.6B), a unified model for text-to-image synthesis and image editing. To enable DPO finetuning on Ovis-U1, we construct a mixed preference corpus that integrates text-to-image and editing pairs, totaling about 33K pairs.

**Training Details and Baselines.** We integrate SDPO into Diffusion-DPO Wallace et al. (2024), DSPO Zhu et al. (2025), and DMPO Li et al. (2025a) implementations and keep their official hyper-parameters. All models are finetuned for 2000 steps with a global batch size of 2048. The learning

Table 1: Reward score comparison on the HPS V2 with SD 1.5. Rows labeled "+ SDPO" report the performance obtained by applying our SDPO to the corresponding base method in the preceding row. $^{\dagger}$: results from our implementation due to the lack of official code. Best results are in **bold**. Owing to space constraints, the full table is provided in the appendix (Table 6).

| Method | PickScore(↑) | HPS(↑) | Aesthetics(↑) | CLIP(↑) | Image Reward(↑) |
|---|---|---|---|---|---|
| SD 1.5 | 0.2088 | 0.2697 | 5.4933 | 0.3480 | -0.0469 |
| SFT | 0.2168 | 0.2838 | 5.7851 | 0.3591 | 0.6619 |
| Diff.-KTO | 0.2164 | 0.2766 | 5.6288 | 0.3420 | 0.5593 |
| MaPO$^{\dagger}$ | 0.2124 | 0.2760 | 5.6890 | 0.3528 | 0.3308 |
| DPOP$^{\dagger}$ | 0.2144 | 0.2780 | 5.7071 | 0.3563 | 0.3735 |
| Diff.-DPO | 0.2131 | 0.2743 | 5.6639 | 0.3552 | 0.1705 |
| + SDPO | 0.2174 | 0.2827 | **5.8744** | 0.3600 | 0.6211 |
| DSPO | 0.2168 | 0.2837 | 5.8346 | 0.3598 | 0.6483 |
| + SDPO | 0.2172 | 0.2847 | 5.8474 | 0.3586 | 0.6578 |
| DMPO$^{\dagger}$ | 0.2131 | 0.2766 | 5.6538 | 0.3551 | 0.3171 |
| + SDPO | **0.2182** | **0.2848** | 5.8574 | **0.3612** | **0.7061** |

Table 2: Average win rate comparison (%) over the HPS V2 using SD 1.5. Each row reports *Model 1* vs. *Model 2* on identical prompts. The upper block summarizes SDPO augmentation results (base + SDPO vs. base), and the lower block compares each model against SD 1.5. Values $> 50\%$ indicate that *Model 1* generally outperforms *Model 2*.

| Model 1 | Model 2 | Pick | HPS V2 | Aesth. | CLIP | ImageReward | Mean |
|---|---|---|---|---|---|---|---|
| *SDPO augmentation effect (base+SDPO vs base)* | | | | | | | |
| Diff.-DPO+ SDPO | Diff.-DPO | 66.12 | 78.62 | **70.62** | 52.50 | **71.62** | 67.90 |
| DSPO + SDPO | DSPO | 52.62 | 53.00 | 53.25 | 48.50 | 52.38 | 51.95 |
| DMPO + SDPO | DMPO | **73.50** | **79.25** | 67.12 | **53.12** | 71.50 | **68.90** |
| *Versus SD 1.5* | | | | | | | |
| Diff.-DPO | SD 1.5 | 76.38 | 69.50 | 66.88 | 57.50 | 63.50 | 66.75 |
| Diff.-DPO+ SDPO | SD 1.5 | 80.75 | 86.25 | **83.38** | 56.88 | 79.25 | 77.30 |
| DSPO | SD 1.5 | 78.38 | 86.00 | 78.75 | **58.88** | 79.75 | 76.35 |
| DSPO + SDPO | SD 1.5 | 81.75 | **89.75** | 79.12 | 56.75 | 80.12 | 77.50 |
| DMPO | SD 1.5 | 68.50 | 74.50 | 68.38 | 53.00 | 69.88 | 66.85 |
| DMPO + SDPO | SD 1.5 | **82.25** | 88.12 | 79.25 | 58.38 | **81.75** | **77.95** |

rate is $1 \times 10^{-8}$ for SD1.5 and $1 \times 10^{-9}$ for SDXL. For the safeguard coefficient $\mu$, on SD 1.5 we set 0.9 for Diffusion-DPO+SDPO and DMPO+SDPO, 0.2 for DSPO+SDPO. On SDXL, $\mu$ is fixed as 0.6 for all variants. We compare against several baselines: the original pretrained SD1.5 and SDXL, supervised finetuning (SFT), Diffusion-KTO Li et al. (2025b), MaPO Hong et al. (2025), DPOP Pal et al. (2024), and original Diffusion-DPO, DSPO, DMPO. For baselines we follow a strict hierarchy. If official checkpoints are publicly available, we evaluate those directly. If checkpoints are unavailable but official code exists, we run the released implementation with the authors' recommended settings. If neither is available, we reimplement the method from the paper.

**Evaluation.** We evaluate models on automatic preference metrics, including PickScore Kirstain et al. (2023), HPS V2 Wu et al. (2023), LAION Aesthetic Classifier Schuhmann et al. (2022), CLIP Radford et al. (2021) and ImageReward Xu et al. (2024) scores. Sampling uses a guidance scale of 7.5 and 50 denoising steps. For Ovis-U1, we additionally evaluate structured text-to-image alignment on GenEval Ghosh et al. (2023) and DPG-Bench Hu et al. (2024), as well as image-editing performance on ImgEdit Ye et al. (2025) and GEdit-EN Liu et al. (2025).

## 5.2 MAIN RESULTS

Table 1 shows that augmenting Diffusion-DPO, DSPO, and DMPO with SDPO consistently improves automatic reward metrics under SD 1.5, with DMPO+SDPO typically achieving the best overall scores. Win-rate results on SD 1.5 (Table 2) further confirm that each base method benefits from SDPO and that SDPO variants also outperform the SD 1.5 baseline, indicating stronger preference alignment without quality loss. On SDXL (Table 3), gains are more moderate yet consistent,

Table 3: Average win rate comparison (%) over the HPS V2 using SDXL.

| Model 1 | Model 2 | Pick | HPS V2 | Aesth. | CLIP | ImageReward | Mean |
|---|---|---|---|---|---|---|---|
| *SDPO augmentation effect (base+SDPO vs base)* | | | | | | | |
| Diff.-DPO+ SDPO | Diff.-DPO | **64.62** | 53.37 | 47.75 | **57.75** | 51.88 | 55.08 |
| DSPO + SDPO | DSPO | 64.25 | **62.62** | **59.75** | 48.62 | **58.13** | **58.67** |
| DMPO + SDPO | DMPO | 55.88 | 53.75 | 52.25 | 55.50 | 56.00 | 54.68 |
| *Versus SDXL* | | | | | | | |
| Diff.-DPO | SDXL | 48.25 | 63.38 | **52.50** | 48.38 | 59.13 | 54.33 |
| Diff.-DPO+ SDPO | SDXL | 59.25 | 66.75 | 49.12 | 55.00 | 58.00 | 57.63 |
| DSPO | SDXL | 37.88 | 57.00 | 40.25 | 55.63 | 56.87 | 49.53 |
| DSPO + SDPO | SDXL | 54.63 | 68.00 | 44.62 | 55.00 | 65.12 | 57.47 |
| DMPO | SDXL | 58.63 | 64.88 | 46.00 | 53.62 | 62.00 | 57.03 |
| DMPO + SDPO | SDXL | **60.88** | **68.50** | 47.63 | **58.38** | **68.63** | **60.80** |

Table 4: Comparison of Ovis-U1 Wang et al. (2025a) variants on preference, structured alignment, and image editing benchmarks. Higher is better (↑). SDPO is particularly effective for preference alignment in large-scale models.

| Model | Preference Eval (↑) | | Structured Alignment Eval (↑) | | Image Editing (↑) | |
|---|---|---|---|---|---|---|
| | CLIP | HPS V2 | GenEval | DPG-Bench | ImgEdit | GEdit-EN |
| Ovis-U1 | 0.3188 | 0.2986 | 0.89 | 83.72 | 4.00 | 6.42 |
| Ovis-U1 + DPO | 0.3192 | 0.2997 | 0.88 | 83.78 | 4.01 | 6.43 |
| Ovis-U1 + SDPO | **0.3201** | **0.3082** | **0.89** | **84.84** | **4.11** | **6.60** |

suggesting reliable scaling to larger backbones. Finally, on the unified Ovis-U1 model (Table 4), SDPO yields clear improvements in preference metrics and editing scores, highlighting effectiveness for large-scale unified generation and editing. Fig. 2 compares SD 1.5 and aligned variants. While naive Diffusion-DPO can enlarge preference margins at the expense of fidelity, our safeguarded integrations preserve details and improve prompt adherence across diverse prompts. The visual evidence aligns with the quantitative trends, indicating that SDPO stabilizes optimization and enhances perceptual quality. See Sec. D for more results.

## 5.3 ABLATION STUDY

**Modular Ablation.** Table 5 compares winner–preserving strategies embedded in MaPO, DPOP, and our SDPO. MaPO introduces a fixed-weight winner loss but omits the reference model, undermining calibration of absolute error. DPOP protects the winner via thresholded update filtering, but its design for autoregressive LLMs creates a modality gap for diffusion models. SDPO addresses both issues by preserving the winner through a safeguard that

Table 5: Ablation study on winner-preserving rules. Model: SD 1.5. Prompts: HPS V2. ‡: fixed $\lambda_{\text{safe}}$ is used in SDPO.

| Method | PickScore (↑) | HPS V2 (↑) |
|---|---|---|
| MaPO | 0.2124 | 0.2760 |
| DPOP | 0.2144 | 0.2780 |
| Diff.-DPO+SDPO‡ | 0.2158 | 0.2803 |
| Diff.-DPO+SDPO | 0.2174 | 0.2827 |

rescales the loser update by $\lambda_{\text{safe}}$ chosen from the output space based on directional alignment. The fixed–$\lambda_{\text{safe}}$ variant already improves over MaPO and DPOP, and the dynamic choice yields further gains. These improvements support the hypothesis that output–space selection of $\lambda_{\text{safe}}$ stabilizes the winner while maintaining pressure to enlarge the preference margin.

**Why does SDPO generalize across DPO variants?** Fig. 3 contrasts the training dynamics of Diff.-DPO, DSPO, and DMPO with or without SDPO. Without SDPO, $\mathcal{L}^w - \mathcal{L}^l$ decreases as expected, whereas $\mathcal{L}^w$ remains nondecreasing and drifts upward in Diff.-DPO and DMPO, indicating unstable optimization. With SDPO, $\mathcal{L}^w$ drops early and remains low, $\mathcal{L}^l$ declines smoothly without overshoot, and $\mathcal{L}^w - \mathcal{L}^l$ decreases steadily to a plateau. We observe a shared qualitative profile across the three SDPO-augmented settings: after basic rescaling, trajectories from different objectives largely overlap. $\mathcal{L}^w$ follows a monotone, fast-then-slow descent, $\mathcal{L}^l$ descends smoothly, and their gap grows in a stable manner across timesteps. This empirical regularity suggests that SDPO successfully corrects harmful update directions and magnitudes by acting on gradient geom-

SD 1.5    Diff.-DPO    Diff.-DPO + SDPO    DSPO    DSPO + SDPO    DMPO    DMPO + SDPO

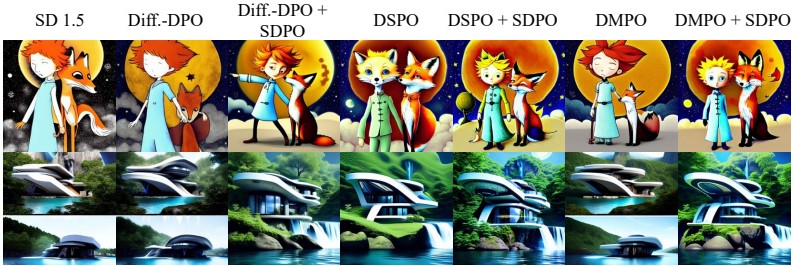

Figure 2: Qualitative comparison of different methods using SD 1.5. Prompt: *1) The Little Prince and the fox in a Tim Burton style artwork. 2) A futuristic modern house on a floating rock island surrounded by waterfalls, moons, and stars on an alien planet.* See Sec. D for more results.

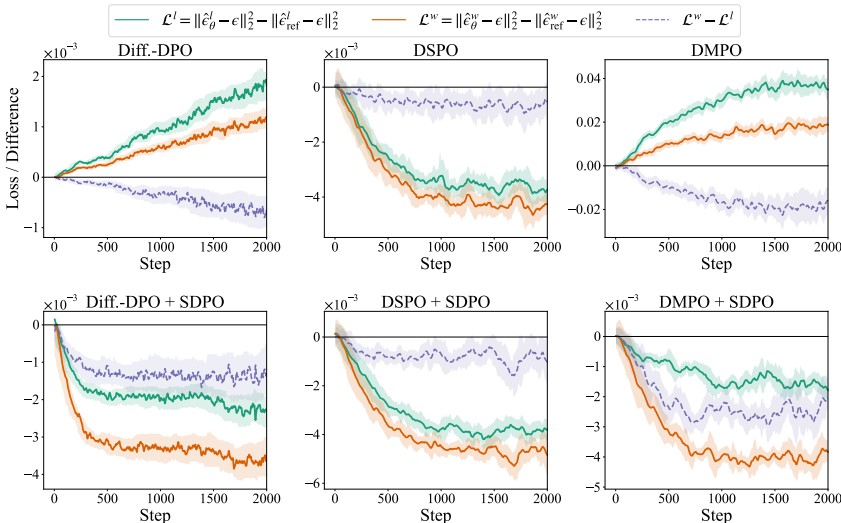

Figure 3: Training dynamics across three objectives with and without SDPO on SD 1.5.

etry rather than on a particular objective form, thereby normalizing training dynamics across DPO variants, preserving the preferred branch, and stabilizing preference alignment.

**Sensitivity of Hyperparameters.** Fig. 4 shows that both HPS V2 and PickScore vary smoothly with $\mu$ and peak at moderate values, indicating low sensitivity. On SDXL, all SDPO-augmented objectives achieve near-optimal performance around $\mu \approx 0.6$ with flat neighborhoods. On SD 1.5, Diffusion-DPO + SDPO and DMPO + SDPO peak near $\mu \approx 0.9$, while DSPO + SDPO peaks near $\mu \approx 0.2$. This aligns with Fig. 3, where DSPO already shows stable decreases in $\mathcal{L}^w$ and $\mathcal{L}^l$, requiring only a small safeguard. Therefore, by default, $\mu$ is set to 0.6 on SDXL for all objectives, while on SD 1.5 it is 0.9 for Diffusion-DPO/DMPO and 0.2 for DSPO.

## 6  CONCLUSIONS AND LIMITATIONS

In this paper, we presented Diffusion-SDPO, a safeguarded preference optimization scheme that stabilizes DPO-style diffusion finetuning by preserving the preferred branch while improving preference matching. The method scales the loser gradient by its alignment with the winner and guarantees, to first order, that the winner's reconstruction loss does not increase. Across SD 1.5, SDXL, and Ovis-U1, our method yields consistent improvements on automated preference, aesthetic, and prompt-alignment metrics with negligible computational overhead, while remaining model-agnostic, straightforward to implement, and applicable to multiple DPO variants. However, the guarantee holds only at first order, so strong curvature in the loss landscape degrades it. Future work includes second-order or trust-region safeguards, automatic or data-driven tuning of $\mu$, extensions to autoregressive preference settings.

## REPRODUCIBILITY STATEMENT

All datasets and base models used in this work are publicly available except for a small mixed preference set that we curate from publicly accessible sources and use only for the Ovis-U1 experiments. Sec. 5.1 details the models, datasets, as well as the complete training and evaluation protocols. Sec. 5.3 reports ablations and sensitivity analyses. Core results on SD 1.5 and SDXL rely solely on public data and base models. The unified Ovis-U1 model exhibits empirical trends consistent with results from SD 1.5 and SDXL. With the documented settings, hyperparameters, and evaluation procedures, independent researchers can reproduce the SD 1.5 and SDXL results using public resources.

## ETHICS STATEMENT

We adhere to the ICLR Code of Ethics. Our experiments rely primarily on public datasets and model releases. For one unified model variant, we curated a small mixed preference set from publicly accessible sources and applied basic filtering for unsafe content where practicable. No human-subjects studies were conducted and, to our knowledge, no personally identifiable information was processed, so institutional review was not required.

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

APPENDIX

## A JUSTIFICATION OF THE NEAR–ISOMETRY

In this section, we provide a detailed justification of the near-isometry property $J^w J^{l^\top} \approx I$. We write the model output at diffusion step $t$ as $o = \epsilon_\theta(x_t, c, t)$. For the winner and loser branches we denote

$$o^w = \epsilon_\theta(x_t^w, c, t), \qquad o^l = \epsilon_\theta(x_t^l, c, t), \qquad J^w = \frac{\partial o^w}{\partial \theta}, \qquad J^l = \frac{\partial o^l}{\partial \theta}. \tag{18}$$

By first–order linearization at the same parameter $\theta$,

$$\nabla_\theta \mathcal{L}^w = J^{w\top} \nabla_o \mathcal{L}^w, \qquad \nabla_\theta \mathcal{L}^l = J^{l\top} \nabla_o \mathcal{L}^l. \tag{19}$$

Hence the parameter–space inner product factorizes as

$$\nabla_\theta \mathcal{L}^{w\top} \nabla_\theta \mathcal{L}^l = \nabla_o \mathcal{L}^{w\top} \underbrace{(J^w J^{l^\top})}_{\text{geometric term}} \nabla_o \mathcal{L}^l. \tag{20}$$

We claim that in our setting $J^w J^{l^\top} \approx I$, which makes the parameter–space geometry well approximated by the Euclidean geometry in output space. Below we state the assumptions and a quantitative bound that together justify this approximation.

**Assumption A (branch proximity).** The two branches are evaluated under the same condition $(c, t)$. For a fixed $t$, the noised latents $x_t^w$ and $x_t^l$ lie in a small neighborhood in latent space. The UNet Ronneberger et al. (2015) is Lipschitz in its input around that neighborhood, therefore the Jacobians are close:

$$\|J^l - J^w\| \leq \varepsilon_{\text{br}}, \qquad \varepsilon_{\text{br}} \ll 1. \tag{21}$$

This follows from sharing the text condition, the timestep, and the architecture, which produce similar intermediate activations and hence similar sensitivities to parameters.

**Assumption B (single–branch near–isometry).** Each branch is locally well conditioned. Empirically, modern diffusion UNets use normalization layers, residual connections, and near unit–gain initializations. Besides, they are trained to predict isotropic Gaussian noise at each step. These design choices constrain the singular values of $J$ to stay close to one. We model this as

$$\|J^w J^{w\top} - I\| \leq \varepsilon_{\text{iso}}, \qquad \|J^l J^{l^\top} - I\| \leq \varepsilon_{\text{iso}}, \tag{22}$$

with $\varepsilon_{\text{iso}} \ll 1$. In words, the map $\theta \mapsto o$ is locally near–isometric.

**A perturbation bound.** Let $E_{\text{br}} := J^l - J^w$ and $E_{\text{iso}} := J^w J^{w\top} - I$. Then

$$J^w J^{l^\top} - I = J^w (J^w + E_{\text{br}})^\top - I = \underbrace{(J^w J^{w\top} - I)}_{E_{\text{iso}}} + J^w E_{\text{br}}^\top. \tag{23}$$

Taking operator norms and using $\|AB\| \leq \|A\| \|B\|$ gives

$$\|J^w J^{l^\top} - I\| \leq \|E_{\text{iso}}\| + \|J^w\| \|E_{\text{br}}\| \leq \varepsilon_{\text{iso}} + (1 + \varepsilon_{\text{gain}}) \varepsilon_{\text{br}}, \tag{24}$$

where $\varepsilon_{\text{gain}} \geq 0$ is defined by $\|J^w\| \leq 1 + \varepsilon_{\text{gain}}$. If both branches are individually near–isometric and their Jacobians are close, then the cross term $J^w J^{l^\top}$ is close to the identity.

**Consequence for gradient geometry.** Combining Eq. 20 and 24 yields

$$\left\| \nabla_\theta \mathcal{L}^{w\top} \nabla_\theta \mathcal{L}^l - \nabla_o \mathcal{L}^{w\top} \nabla_o \mathcal{L}^l \right\| \leq \|\nabla_o \mathcal{L}^w\| \|\nabla_o \mathcal{L}^l\| \cdot \|J^w J^{l^\top} - I\|, \tag{25}$$

so the inner product between parameter–space gradients is close to the inner product between output–space gradients. Intuitively, in the small neighborhood induced by the shared $(c, t)$, the network behaves almost angle– and length–preserving along the two branches, which justifies replacing the parameter–space inner product by the Euclidean dot product in output space.

**Why Assumption B is plausible for diffusion UNets.** At a fixed $t$, the training objective is a mean–squared error between the model output and an isotropic Gaussian target. This induces a whitened curvature in output space. Together with normalization layers and residual pathways, which are known to promote dynamical isometry by controlling singular values Ioffe & Szegedy (2015); Tarnowski et al. (2019), the per–step Jacobian spectrum concentrates around one. This effect is stable under small finetuning steps and within the local region defined by $x_t^w$ and $x_t^l$.

**Empirical diagnostics.** We validate the approximation by comparing the cosine between parameter space gradients $\rho_\theta = \cos\left(\nabla_\theta \mathcal{L}^w, \nabla_\theta \mathcal{L}^l\right)$ with the cosine between output space gradients $\rho_o = \cos\left(\nabla_o \mathcal{L}^w, \nabla_o \mathcal{L}^l\right)$ across timesteps and prompts. Steps with near zero norms are filtered to avoid numerical artifacts. We summarize the absolute gap $\Delta\rho_t = \left|\rho_\theta(t) - \rho_o(t)\right|$ and observe small values of $\Delta\rho_t$ for the majority of steps, which supports Eq. 20 with $J^w {J^l}^\top \approx I$.

## B  SECOND-ORDER CONSIDERATIONS OF SDPO

It is important to note that our guarantee is based on a first-order (linear) approximation of the loss landscape. In reality, the true change in $\mathcal{L}^w$ after an update includes higher-order terms: $\Delta\mathcal{L}^w = \nabla_\theta {\mathcal{L}^w}^\top \Delta\theta + \frac{1}{2}\Delta\theta^\top H^w \Delta\theta + \mathcal{O}(|\Delta\theta|^3)$, where $H^w$ is the Hessian of $\mathcal{L}^w$. Diffusion-SDPO does not explicitly account for the $\frac{1}{2}\Delta\theta^\top H^w \Delta\theta$ term. If the curvature (eigenvalues of $H^w$) is large or the step $\Delta\theta$ is not infinitesimally small, it is conceivable that $\mathcal{L}^w$ could increase slightly even when the first-order term is zero or negative.

To hedge against curvature, we adopt a margin parameter $\mu \in [0, 1]$ and set $\lambda_{\text{safe}} \leftarrow (1 - \mu)\,\lambda_{\text{safe}}$ (e.g., $\mu = 0.1$ gives $0.9\,\lambda_{\text{safe}}$). This adjustment helps ensure $\Delta\mathcal{L}^w$ stays negative even when small second-order effects are present. Empirically, we found that the first-order proxy combined with this margin was adequate for stability since the second-order error term remained small relative to the linear term in our experiments.

Another related source of approximation error is the assumption that $J^w {J^l}^\top \approx I$ in Eq. 15. In pathological cases where the model's parameterization leads to very different Jacobians for $x_0^w$ and $x_0^l$, the output-space dot product $g^{w\top} g^l$ might not perfectly predict the sign of the parameter-space dot product. In practice, however, we expect $g^{w\top} g^l$ to be a reliable indicator of gradient alignment, since $g^w$ and $g^l$ live in the same output vector space and capture intuitive per-pixel error correlations between the two samples (see Sec. A for more evidence on this assumption). Our safe update rule can thus be viewed as a principled, efficient heuristic that steers the optimization to avoid obvious "bad" directions that would hurt the preferred outcome. By explicitly safeguarding $\mathcal{L}^w$ from increases, Diffusion-SDPO favors a more conservative yet effective alignment, in contrast to methods that risk overshooting in pursuit of preference satisfaction. Generally, our approach leads to a more stable training process that maintains or even improves the quality of winner outputs while still driving down the relative loss of loser outputs, ultimately yielding a well-aligned diffusion model.

## C  LLM USAGE DISCLOSURE

We used the large language model (LLM) only to aid English writing quality, including grammar correction, style polishing, and minor rephrasing at the sentence or paragraph level. The LLM did not generate research ideas, algorithms, proofs, datasets, code, experimental designs, figures, tables, statistical analyses, or results. All technical content, claims, and conclusions were created and verified by the authors. To reduce risk of factual errors, every LLM-suggested edit was reviewed by at least one author and cross-checked against our methods, experiments, and citations. No undisclosed prompts, hidden instructions, or external links intended to influence the review process were included in the submission. The authors take full responsibility for all content in this paper, including any text edited with LLM assistance, and the LLM is not an author or contributor.

Table 6: Full results of reward score comparison on Pick-a-Pic V2, HPS V2, and PartiPrompts using SD 1.5. $^\dagger$: results from our implementation due to the lack of official code.

| Dataset | Method | PickScore(↑) | HPS(↑) | Aesthetics(↑) | CLIP(↑) | Image Reward(↑) |
|---|---|---|---|---|---|---|
| Pick V2 | SD 1.5 | 0.2073 | 0.2651 | 5.3907 | 0.3299 | -0.1376 |
| | SFT | 0.2128 | 0.2765 | 5.6888 | 0.3408 | 0.5767 |
| | Diff.-KTO | 0.2126 | 0.2766 | 5.6288 | 0.3420 | 0.5593 |
| | MaPO$^\dagger$ | 0.2097 | 0.2702 | 5.5572 | 0.3365 | 0.2435 |
| | DPOP$^\dagger$ | 0.2119 | 0.2726 | 5.5688 | 0.3389 | 0.3259 |
| | Diff.-DPO | 0.2109 | 0.2690 | 5.4958 | 0.3357 | 0.1020 |
| | + **SDPO** | 0.2143 | 0.2772 | 5.7172 | 0.3458 | 0.5546 |
| | DSPO | 0.2131 | 0.2769 | 5.6825 | 0.3428 | 0.5642 |
| | + **SDPO** | 0.2135 | 0.2777 | 5.6917 | 0.3441 | 0.5916 |
| | DMPO$^\dagger$ | 0.2110 | 0.2710 | 5.5434 | 0.3382 | 0.2813 |
| | + **SDPO** | **0.2144** | **0.2784** | **5.7312** | **0.3469** | **0.6369** |
| HPS V2 | SD 1.5 | 0.2088 | 0.2697 | 5.4933 | 0.3480 | -0.0469 |
| | SFT | 0.2168 | 0.2838 | 5.7851 | 0.3591 | 0.6619 |
| | Diff.-KTO | 0.2164 | 0.2766 | 5.6288 | 0.3420 | 0.5593 |
| | MaPO$^\dagger$ | 0.2124 | 0.2760 | 5.6890 | 0.3528 | 0.3308 |
| | DPOP$^\dagger$ | 0.2144 | 0.2780 | 5.7071 | 0.3563 | 0.3735 |
| | Diff.-DPO | 0.2131 | 0.2743 | 5.6639 | 0.3552 | 0.1705 |
| | + **SDPO** | 0.2174 | 0.2827 | **5.8744** | 0.3600 | 0.6211 |
| | DSPO | 0.2168 | 0.2837 | 5.8346 | 0.3598 | 0.6483 |
| | + **SDPO** | 0.2172 | 0.2847 | 5.8474 | 0.3586 | 0.6578 |
| | DMPO$^\dagger$ | 0.2131 | 0.2766 | 5.6538 | 0.3551 | 0.3171 |
| | + **SDPO** | **0.2182** | **0.2848** | 5.8574 | **0.3612** | **0.7061** |
| PartiPrompts | SD 1.5 | 0.2144 | 0.2724 | 5.3466 | 0.3343 | 0.0637 |
| | SFT | 0.2181 | 0.2821 | 5.5981 | 0.3389 | 0.5830 |
| | Diff.-KTO | 0.2178 | 0.2820 | 5.5630 | 0.3416 | 0.5697 |
| | MaPO$^\dagger$ | 0.2152 | 0.2754 | 5.4754 | 0.3366 | 0.3358 |
| | DPOP$^\dagger$ | 0.2169 | 0.2782 | 5.4894 | 0.3383 | 0.3644 |
| | Diff.-DPO | 0.2167 | 0.2755 | 5.4045 | 0.3391 | 0.2560 |
| | + **SDPO** | 0.2187 | 0.2815 | 5.5880 | 0.3423 | 0.5425 |
| | DSPO | 0.2178 | 0.2819 | **5.5997** | 0.3385 | 0.5640 |
| | + **SDPO** | 0.2185 | **0.2832** | 5.5975 | 0.3405 | 0.5955 |
| | DMPO$^\dagger$ | 0.2163 | 0.2775 | 5.4724 | 0.3388 | 0.3653 |
| | + **SDPO** | **0.2190** | 0.2831 | 5.5956 | **0.3430** | **0.6381** |

# D    FULL EXPERIMENTAL RESULTS

We report the full reward score comparisons on Pick-a-Pic V2, HPS V2, and PartiPrompts in Tables 6 and 7. Figure 4 analyzes the sensitivity of $\mu$, the only hyperparameter introduced by our method, by plotting HPS V2 and PickScore as functions of $\mu$. We also provide additional qualitative results in Fig. 5 and 6. Taken together, these quantitative and qualitative results indicate that SDPO consistently improves preference alignment while maintaining, and in many cases enhancing, visual quality.

Table 7: Full results of reward score comparison on Pick-a-Pic V2, HPS V2, and PartiPrompts using SDXL. [†]: results from our implementation due to the lack of official code.

| Dataset | Method | PickScore($\uparrow$) | HPS($\uparrow$) | Aesthetics($\uparrow$) | CLIP($\uparrow$) | Image Reward($\uparrow$) |
|---|---|---|---|---|---|---|
| Pick V2 | SDXL | 0.2242 | 0.2846 | 5.9970 | 0.3684 | 0.7382 |
| | SFT | 0.2183 | 0.2809 | 5.7922 | 0.3658 | 0.5974 |
| | MaPO | 0.2242 | 0.2871 | **6.0979** | 0.3684 | 0.8359 |
| | Diff.-DPO | 0.2251 | 0.2868 | 6.0115 | 0.3732 | 0.8357 |
| | + **SDPO** | 0.2257 | 0.2876 | 5.9812 | 0.3746 | 0.8840 |
| | DSPO | 0.2228 | 0.2834 | 5.8797 | 0.3756 | 0.8818 |
| | + **SDPO** | 0.2240 | 0.2871 | 5.9529 | 0.3761 | 0.9238 |
| | DMPO[†] | 0.2253 | 0.2869 | 6.0119 | 0.3716 | 0.8555 |
| | + **SDPO** | **0.2263** | **0.2882** | 5.9990 | **0.3770** | **0.9548** |
| HPS V2 | SDXL | 0.2290 | 0.2900 | 6.1271 | 0.3847 | 0.9047 |
| | SFT | 0.2228 | 0.2883 | 5.9689 | 0.3806 | 0.8528 |
| | MaPO | 0.2293 | 0.2934 | **6.1882** | 0.3840 | 0.9703 |
| | Diff.-DPO | 0.2288 | 0.2927 | 6.1380 | 0.3840 | 1.0159 |
| | + **SDPO** | 0.2308 | 0.2938 | 6.1284 | 0.3879 | 1.0326 |
| | DSPO | 0.2273 | 0.2916 | 6.0424 | 0.3894 | 1.0054 |
| | + **SDPO** | 0.2293 | **0.2944** | 6.1040 | 0.3889 | **1.0745** |
| | DMPO[†] | 0.2302 | 0.2921 | 6.1101 | 0.3875 | 1.0154 |
| | + **SDPO** | **0.2308** | 0.2933 | 6.1113 | **0.3897** | 1.0521 |
| PartiPrompts | SDXL | 0.2277 | 0.2880 | 5.7901 | 0.3591 | 0.8573 |
| | SFT | 0.2221 | 0.2834 | 5.6496 | 0.3559 | 0.7515 |
| | MaPO | 0.2278 | 0.2902 | **5.8921** | 0.3580 | 0.9324 |
| | Diff.-DPO | 0.2279 | 0.2900 | 5.8294 | 0.3629 | 1.0638 |
| | + **SDPO** | 0.2290 | 0.2907 | 5.7882 | 0.3645 | 1.0654 |
| | DSPO | 0.2261 | 0.2871 | 5.6947 | 0.3664 | 1.0514 |
| | + **SDPO** | 0.2268 | 0.2897 | 5.7931 | **0.3664** | **1.1012** |
| | DMPO[†] | 0.2286 | 0.2904 | 5.8273 | 0.3610 | 0.9558 |
| | + **SDPO** | **0.2296** | **0.2913** | 5.8103 | 0.3649 | 1.0623 |

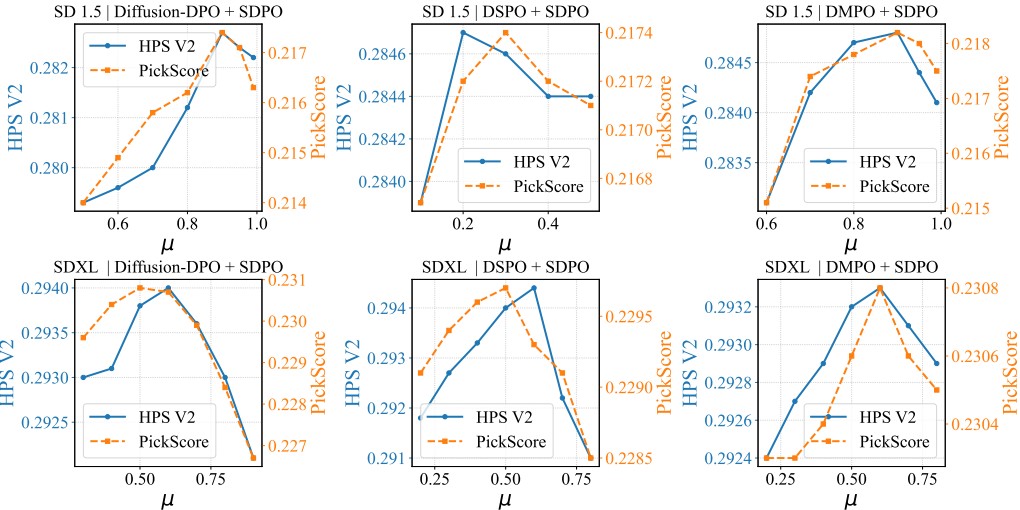

Figure 4: Sensitivity of SDPO to the hyperparameter $\mu$ measured by HPS V2 and PickScore across SD 1.5 and SDXL on HPS V2 prompt set.

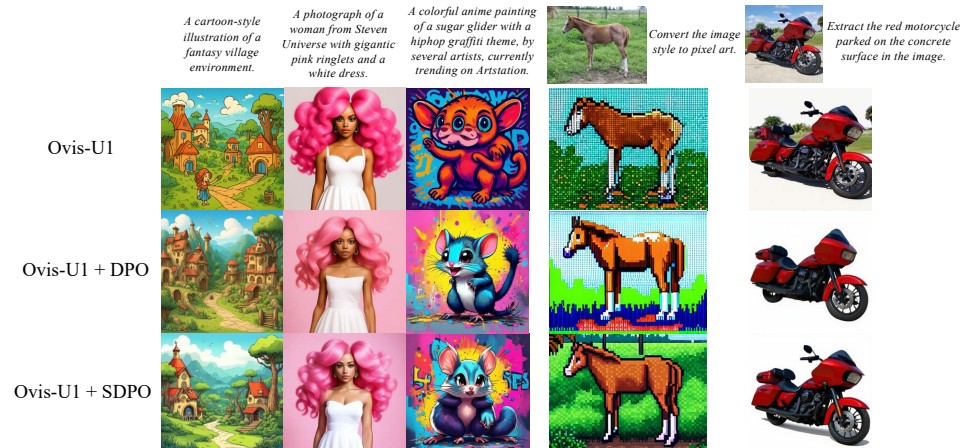

Figure 5: Qualitative comparison of images generated by Ovis-U1 Wang et al. (2025a) and its fine-tuned variants. Results are reported for three variants: the base, finetuned with DPO, and finetuned with our SDPO.

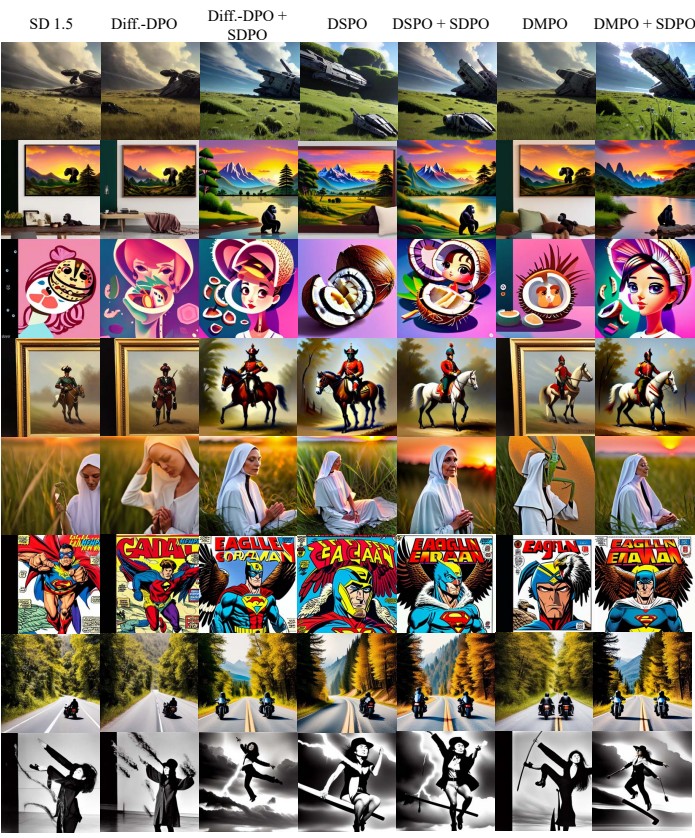

Figure 6: Qualitative comparison of images generated by different methods using SD 1.5. Prompt: *1) A hyper-realistic landscape from a Neil Blomkamp film featuring a crashed spaceship, detailed grass, and a photorealistic sky. 2) A landscape featuring mountains, a valley, sunset light, wildlife and a gorilla, reminiscent of Bob Ross's artwork. 3) A stylized portrait featuring sliced coconut, electronics, and AI in a cartoonish cute setting with a dramatic atmosphere. 4) A tonalist painting of a bipedal pony creature soldier. 5) A praying mantis nun in a grassy field during sunset. 6) A comic book cover featuring a superhero named "Eagle Man" with an eagle mask and wing logo, resembling a traditional comic book cover. 7) Two motorcycles sit on the side of a secluded road. 8) Yoko Ono flying on a broomstick with lightning in the skies.*

