# OpenReview forum: "Diffusion-SDPO: Safeguarded Direct Preference Optimization for Diffusion Models"
_ICLR.cc/2026/Conference — ICLR 2026 Conference Withdrawn Submission_

### Official Review · Reviewer_Ad82 · 2025-10-28

**Soundness:** 3
**Presentation:** 2
**Contribution:** 3
**Rating:** 4
**Confidence:** 5

**Summary:**

The authors identify a failure mode of the classical DPO objective: both winner and loser losses tend to increase during training, which is highly counterintuitive. Using a first-order Taylor expansion, they derive a simple condition that ensures the winner’s loss does not increase. Under the assumption that the product of the winner’s and loser’s Jacobians is approximately an identity matrix, they introduce a dynamic scaling coefficient that adaptively balances the loser’s contribution, guaranteeing (under the linearization and Jacobian assumptions) non-increasing winner loss. Empirically, the method achieves strong results across standard synthetic reward models, consistently outperforming baseline alignment approaches with a simple, plug-and-play modification.

**Strengths:**

1. The proposed method is simple, elegant, and practical — a few-line modification that can be readily integrated into existing DPO pipelines.

2. The paper identifies and addresses a highly counterintuitive failure mode of classical DPO, providing clear analytical insight and an effective solution.

3. The theoretical assumptions are reasonable, and the mathematical analysis is clear and easy.

**Weaknesses:**

1. While the assumptions are generally realistic, the analysis relies heavily on specific properties of U-Nets. This limits the generality of the results, as U-Nets are already completely replaced by more modern architectures such as Diffusion Transformers (DiTs). The paper would be significantly stronger if the authors discussed how their assumptions extend to DiTs or at least provided quantitative results on such architectures.

2. The paper lacks an analysis of how the dynamic λ coefficients evolve during training. Visualizations or comparisons with fixed λ baselines would help illustrate the mechanism’s behavior and further validate the claimed improvements.

**Questions:**

1. Can the proposed approach be supported by weaker or more general assumptions about the Jacobians? Alternatively, could you provide results or analysis using Diffusion Transformers (DiTs) to demonstrate that the method generalizes beyond U-Nets? Any clarification or quantitative evidence in this direction would be very helpful.

2. Given the practical nature of this work, additional ablations would substantially strengthen the paper—for example, comparing against constant λ values or visualizing how λ evolves during training.

I would be happy to raise my score if the authors address these concerns, as the paper is already practical, easy to follow, and offers a genuinely few-line, plug-and-play improvement. I understand the tight rebuttal timeline and would appreciate any additional clarification the authors can provide.

---

### Official Review · Reviewer_Gmbd · 2025-10-30

**Soundness:** 2
**Presentation:** 3
**Contribution:** 3
**Rating:** 4
**Confidence:** 4

**Summary:**

The authors identify a limitation in conventional DPO training that updates can increase the reconstruction error of both the winner and loser branches. To address this, they propose Diffusion-SDPO, a safeguarded update rule that preserves the winner by adaptively scaling the loser’s gradient according to its alignment with the winner’s gradient.

**Strengths:**

1. The paper provides a clear analysis of the shortcomings of standard DPO and argues that SDPO enables more stable optimization during preference training.
2. The mathematical derivations are detailed and carefully presented, supporting the approximate solution procedure for Diffusion-SDPO.

**Weaknesses:**

1. The paper lacks a discussion of the computational overhead of computing $\lambda_{safe}$. For example, there is no comparison with baselines in terms of training memory usage or per-step backward-pass time.
2. All experiments are trained for only 2,000 steps. It is unclear whether performance has reached the peak for each baseline or whether results are reported before full convergence. Longer-horizon training results would help clarify this.
3. From Figure 3, the original DPO variant does not appear to exhibit the property described in the paper (“the objective can increase the reconstruction error of both winner and loser branches,” lines 15–16). The paper does not discuss how SDPO improves performance in this case.

**Questions:**

NA

---

### Official Review · Reviewer_eiGm · 2025-10-30

**Soundness:** 2
**Presentation:** 3
**Contribution:** 2
**Rating:** 4
**Confidence:** 4

**Summary:**

Aligning Text-to-image diffusion models with human aesthetic and preference judgments remains difficult. Existing Direct Preference Optimization (DPO) methods have been adapted to diffusion models to improve preference alignment. However, the authors identify a key flaw in this approach: increasing the preference margin (between “winner” and “loser” samples) does not necessarily improve overall image quality.
In practice, standard Diffusion-DPO often degrades both winner and loser reconstructions, implying that alignment gains come from worsening the loser rather than improving the winner. This leads to unstable training and even collapse in some cases. To address this, the paper introduces Diffusion-SDPO (Safeguarded Direct Preference Optimization) a modification to DPO that preserves the winner sample’s quality during optimization. The method introduces an adaptive scaling of the loser gradient, determined by how aligned it is with the winner gradient. If the loser’s gradient direction conflicts with the winner’s, its contribution is downweighted.

**Strengths:**

1. Insightful Diagnosis of DPO Behavior : The paper makes a clear and valuable observation that simply increasing the preference margin in diffusion-based DPO does not guarantee improved image quality. By identifying that both winner and loser losses can rise during training, the authors uncover a subtle but important failure mode in current preference optimization methods.
2. Effective Solution : The proposed Diffusion-SDPO introduces a simple modification, adaptive scaling of the loser gradient, that yields substantial empirical gains. Despite its conceptual simplicity and minimal computational cost, the method demonstrates consistent improvements across models and benchmarks, highlighting its strong practical impact.

**Weaknesses:**

The theoretical analysis includes several nontrivial leaps that are not fully justified.
In particular, the assumption that $J_w^{\top} J_l = I$ (identity matrix) appears unrealistic and lacks both empirical and conceptual grounding in the context of diffusion model optimization.
Assumptions A and B are introduced without sufficient explanation or validation, making it difficult to assess their plausibility.  For example, in assumption A, "For a fixed t, the noised latents $x_t^w$ and $x_t^l$
 lie in a small neighborhood in latent space". I do not agree with this. In assumption B, "They are trained to predict isotropic Gaussian noise at each step. These design choices constrain the singular values of J to stay close to one." Predicting isotropic noise means that the outputs  have similar variance in all components. I believe it says nothing about how sensitively each output dimension depends on the parameters.

**Questions:**

1. Could you please include a small empirical study (or a clearer derivation) to justify the near-isometry assumption $J_w J_l^{\top} \approx I$. For a toy UNet or a narrow block, estimate Jacobian and report, such as (i) the spectrum of $J J^{\top}$ and (ii) $\|J_w - J_l\|$ when $x_t^w$ and $x_t^l$
are nearby across several timesteps $t$ and prompts. Even low-rank probes would clarify the frequency and extent to which the assumption holds.

2. If the authors’ claim holds, a compelling validation would be to compare SDPO with baseline DPO methods trained for a larger number of steps more than 2000. In such extended training, baseline methods are expected to continue widening the preference margin but often at the cost of visual fidelity, producing “ugly” or degraded images as shown in Figure 1 despite improvements in preference scores.
In contrast, if SDPO effectively constrains the increase of $L_w$, the preferred-image quality should remain stable even with prolonged training. I suggest including both quantitative and qualitative evidence for this effect. Quantitatively, report trends of aesthetic or NR-IQA metrics over extended training steps. Qualitatively, show fixed-seed generations at different checkpoints to visualize whether baseline methods degrade while SDPO maintains perceptual quality.

---

### Official Review · Reviewer_wzME · 2025-11-01

**Soundness:** 2
**Presentation:** 3
**Contribution:** 3
**Rating:** 6
**Confidence:** 3

**Summary:**

The paper addresses an critical issue in Diffusion-DPO, where the model can widen the preference gap by making both outputs worse, including the preferred one. It adds a simple winner-preserving update that checks how the winner and loser gradients align and scales down the loser’s influence when it would hurt the winner. The method plugs into existing DPO-style training with minimal changes and compute. In experiments on popular text-to-image models, it stabilizes training and delivers consistent gains in preference, aesthetic quality, and prompt alignment.

**Strengths:**

1.	The paper identifies and tackles a critical problem in Diffusion-DPO that the optimization can hack the objective by making the lose samples worse than winning samples.

2.	The paper proposes an easy rescaling method based on first-order analysis to maintain the win samples quality.

3.	The proposed method is validated on various models on different benchmarks.

**Weaknesses:**

1.	The key assumption on the near-isometry is fragile. It relies on both near-isometry of self-Jacobian and closeness of Jacobians in two branches, which can break easily. Current analysis is also based on U-Net, which does not include other architectures like DiT.

2.	A more formal analysis on simplifying the DPO objective to a linear version. It’s not quite clear that the current analysis will still hold in putting back to the sigmoid DPO loss.

3.	The performance gain on SDXL over benchmark methods is minimal.

**Questions:**

1.	From SDXL results in Table 6, it can be seen that the proposed method plugged into DMPO consistently achieves a larger increase than plugging into Diff.-DPO and DSPO, why is it this case?

2.	How do the authors apply the proposed method to DSPO, where the L^l and L^w is not directly clear in the their objective.

3.	How does $\mu$ collaborate with $\beta$ in practice?

---

### Author Response · Authors · 2025-11-13

We sincerely thank the reviewers and the area chairs for their valuable time and constructive feedback. Their comments have provided us with important insights that will help us further improve the quality and clarity of our work. After careful consideration, we have decided to withdraw the current submission in order to refine our experiments, strengthen the analysis, and better address the raised concerns. We deeply appreciate the reviewers and area chairs’ efforts.

---

### Note · Authors · 2025-11-13

I have read and agree with the venue's withdrawal policy on behalf of myself and my co-authors.